# Metformin: A Dual-Role Player in Cancer Treatment and Prevention

**DOI:** 10.3390/ijms25074083

**Published:** 2024-04-06

**Authors:** Mariam Ahmed Galal, Mohammed Al-Rimawi, Abdurrahman Hajeer, Huda Dahman, Samhar Alouch, Ahmad Aljada

**Affiliations:** 1Department of Biochemistry and Molecular Medicine, College of Medicine, Alfaisal University, P.O. Box 50927, Riyadh 11533, Saudi Arabia; maelsayed@alumni.alfaisal.edu (M.A.G.); alrimawimohammad1@gmail.com (M.A.-R.); hdahman@alfaisal.edu (H.D.); salouch@alfaisal.edu (S.A.); 2Department of Translational Health Sciences, Bristol Medical School, University of Bristol, Bristol BS8 1QU, UK; 3Medical School, University of Manchester, Manchester M14 4PX, UK; hajeer.abdul@gmail.com

**Keywords:** metformin, cancer, chemoresistance, anticancer, adjuvant therapy, mechanisms of action, clinical trials

## Abstract

Cancer continues to pose a significant global health challenge, as evidenced by the increasing incidence rates and high mortality rates, despite the advancements made in chemotherapy. The emergence of chemoresistance further complicates the effectiveness of treatment. However, there is growing interest in the potential of metformin, a commonly prescribed drug for type 2 diabetes mellitus (T2DM), as an adjuvant chemotherapy agent in cancer treatment. Although the precise mechanism of action of metformin in cancer therapy is not fully understood, it has been found to have pleiotropic effects, including the modulation of metabolic pathways, reduction in inflammation, and the regulation of cellular proliferation. This comprehensive review examines the anticancer properties of metformin, drawing insights from various studies conducted in vitro and in vivo, as well as from clinical trials and observational research. This review discusses the mechanisms of action involving both insulin-dependent and independent pathways, shedding light on the potential of metformin as a therapeutic agent for different types of cancer. Despite promising findings, there are challenges that need to be addressed, such as conflicting outcomes in clinical trials, considerations regarding dosing, and the development of resistance. These challenges highlight the importance of further research to fully harness the therapeutic potential of metformin in cancer treatment. The aims of this review are to provide a contemporary understanding of the role of metformin in cancer therapy and identify areas for future exploration in the pursuit of effective anticancer strategies.

## 1. Background

Globally, cancer remains a significant public health concern, resulting in a staggering number of deaths and new cases. In 2020 alone, there were approximately 10 million deaths and 19.3 million new cases reported [1,2]. Tragically, cancer claims the lives of nearly one in six individuals, with liver, colon, stomach, and breast cancers being particularly fatal, causing 788,000, 774,000, 754,000, and 571,000 deaths, respectively [3]. Disturbingly, the incidence of cancer is projected to rise even further, with Cancer Research UK estimating a 54.9% increase, leading to 28 million new cases by 2040 [4]. This alarming trend can be attributed to various factors, such as aging, hyperlipidemia, type 2 diabetes mellitus (T2DM), and obesity, all of which contribute significantly to the heightened incidence of cancer. Despite the progress made in chemotherapy, the emergence of chemoresistance and relapse poses significant challenges to achieving successful cancer treatment [1].

Metformin, an oral biguanide, is the most recommended drug for T2DM [5]. Despite the fact that its precise mechanism of action remains poorly understood, metformin has been shown to decrease liver gluconeogenesis, enhance insulin sensitivity in muscle and adipose tissue, and increase glucose uptake in the gut [5]. Currently, metformin is well known for its pleiotropic effects, which include reducing body weight, influencing the lipid profile, and regulating inflammatory markers. A notable reduction in the cancer incidence among metformin users has been reported in an observational study [6], leading to a significant interest in exploring metformin’s potential for cancer prevention and treatment. Several mechanisms of action have been proposed for metformin in the context of cancer. One such mechanism involves its ability to reverse the Warburg effect observed in cancer cells, characterized by an increase in fatty acid degradation and anaerobic glycolysis [7,8]. Metformin also exerts its influence on cancer cells by modulating the AMPK/mTOR pathway and inhibiting cytokines [3]. Furthermore, it affects cell proliferation markers [9] and mitigates the production of reactive oxygen species, oxidative stress, and DNA damage [10].

The clinical translation of metformin poses a challenge due to conflicting results between clinical trials and preclinical studies. This comprehensive review aims to provide an updated understanding of metformin’s potential as an anticancer agent. It examines data from in vitro and in vivo studies, exploring the efficacy of metformin as a standalone or adjuvant drug in the treatment of various cancers. A comprehensive literature search identified relevant studies in PubMed and Google Scholar using terms like “metformin and cancer”, “metformin and interventional chemotherapy”, and “metformin mechanism of action”. This ensured the inclusion of in vitro and in vivo studies on metformin’s anticancer effects across diverse cancers. Additionally, relevant epidemiological studies on metformin and cancer risk reduction were incorporated. Finally, information on ongoing clinical trials was obtained from ClinicalTrials.gov (NIH) using “cancer”, “chemotherapy”, and “metformin” for a current understanding of metformin’s potential in cancer prevention and treatment. Any study with incomplete results was excluded. Additionally, this review addresses the controversies surrounding the repurposing of metformin as an anticancer agent and the regulatory aspects to be considered.

## 2. Metformin as an Anticancer Drug: In Vivo and In Vitro Studies

In 2005, Evans et al. first reported the anticancer effects of metformin in their pilot case–control study conducted in Scotland [6]. They found that metformin-treated diabetic patients had a lower incidence of cancer compared to those taking other diabetes medications. Similarly, Bowker et al. demonstrated a significant decrease in the cancer-related mortality among T2DM patients on metformin compared to those on sulfonylureas or insulin [11]. Several observational studies and meta-analyses have consistently reported a remarkable reduction in the incidence of various cancer types and improved outcomes with the use of metformin (Table 1). Interestingly, this effect has been observed in both diabetic and non-diabetic patients [12]. Furthermore, several preclinical studies have been conducted to evaluate the anticancer properties of metformin, including its synergy with cytotoxic chemotherapy and radiotherapy. In vitro studies have demonstrated that metformin sensitizes cells to various chemotherapeutic agents by reducing their 50% inhibitory concentration (IC50) (Table 2). Additionally, numerous studies have aimed to elucidate the pathways involved in these effects. For example, recent research has shown that metformin increases radiosensitivity by influencing the production of reactive oxygen species (ROS) after radiotherapy and by promoting the degradation of HIF-1α, a protein associated with radio-resistance [13]. Similarly, several interventional clinical trials have been conducted to evaluate the role of metformin in cancer patients (Table 3).

However, it is important to note that some of these trials failed to confirm the anticancer role of metformin. The reasons for these discrepancies are discussed later in Section 6 of this review. It is noteworthy to mention that the generalizability of the current evidence is hampered by several methodological shortcomings. For instance, the small sample sizes raise concerns about the robustness of the findings and increase the likelihood of type I and type II errors. While in type I errors, a statistically significant effect is mistakenly observed due to chance alone, type II errors lead to the failure to detect a true effect because the study lacks sufficient statistical power. Second, restricted follow-up durations limit the ability to capture long-term effects. Third, the absence of data on potential confounders, such as obesity, diet, and physical activity, impedes proper adjustment. Fourth, the incomplete reporting of metformin adherence introduces uncertainty regarding the true treatment effect. Finally, susceptibility to time-related biases, including immortal time bias, time-window bias, and time-lag bias [14,15,16,17,18], might inflate the observed protective effects of metformin. Collectively, these limitations suggest potential tumor site- or type-specific benefits of metformin, contributing to the observed inconsistencies across clinical trials.

**Table 1 ijms-25-04083-t001:** Observational studies investigating the association between metformin and cancer incidence.

Authors [Ref.]	Year	Study Design	Outcome	Relative Risk (95% CI) ^a^
Bowker et al. [11]	2006	Cohort	Cancer mortality	0.8 (0.6–0.9)
Currie et al. [19]	2009	Cohort	Any cancer	0.54 (0.43–0.66)
Wright et al. [20]	2009	Case–control	Prostate cancer	0.56 (0.32–1.00)
Libby et al. [21]	2009	Cohort	Any cancer	0.63 (0.53–0.75)
Bowker et al. [22]	2010	Cohort	Cancer mortality	0.8 (0.65–0.98)
Dandon et al. [23]	2010	Case–control	Liver cancer	0.15 (0.04–0.5)
Bodmer et al. [24]	2010	Nested case–control	Breast cancer	0.44 (0.24–0.82)
Lee et al. [25]	2011	Cohort	Any cancer	0.12 (0.08–0.19)
Chen et al. [26]	2011	Cohort	Liver cancer	0.24 (0.07–0.8)
He et al. [27]	2011	Cohort	Prostate: all-cause mortality	0.55 (0.32–0.94)
Monami et al. [28]	2011	Nested case–control	Any cancer	0.28 (0.13–0.57)
Bosco et al. [29]	2011	Nested case–control	Breast cancer	0.81 (0.63–0.96)
Bodmer et al. [30]	2011	Nested case–control	Ovarian cancer	0.61 (0.3–1.25)
Geraldine et al. [31]	2012	Cohort	Any cancer	0.2 (0.03–0.82)
Lai et al. [32]	2012	Cohort	Lung cancer	0.55 (0.32–0.94)
Lee et al. [33]	2012	Cohort	Colorectal: all-cause mortality/cancer mortality	0.66 (0.45–0.98)/0.66 (0.48–0.92)
He et al. [34]	2012	Cohort	Breast: all-cause mortality/cancer mortality	0.52 (0.28–0.97)/0.47 (0.24–0.9)
Romero et al. [35]	2012	Cohort	Ovarian cancer progression/all-cause mortality	0.38 (0.16–0.90)/0.43 (0.16–1.19)
Bodmer et al. [36]	2012	Nested case–control	Pancreatic cancer	0.43 (0.23–0.8)
Bodmer et al. [36]	2012	Nested case–control	Colorectal cancer	1.43 (1.08–1.9)
Franciosi et al. [37]	2013	Systematic review	Any cancer: all-cause mortality	0.65 (0.53–0.80)
Preston et al. [38]	2014	Nested case–control	Prostate cancer	0.84 (0.74–0.96)
Kim et al. [39]	2014	Cohort	Gastric cancer	0.57 (0.37–0.87) ^b^
Chen et al. [40]	2015	Cohort	Any cancer	1.36 (1.11–1.67)
Calip et al. [41]	2016	Cohort	Breast cancer	0.95 (0.51–1.77)
Häggström et al. [42]	2016	Cohort	Prostate cancer	0.96 (0.77–1.19)
Franchi et al. [43]	2017	Nested case–control	Endometrial cancer	0.99 (0.80–1.23)
Kim et al. [44]	2018	Cohort	Any cancer	0.513 (0.318–0.826)
Chang et al. [45]	2018	Cohort	Colorectal cancer	0.36 (0.29–0.44) ^c^, 0.6 (0.49–0.74) ^d^
Tang et al. [46]	2018	Meta-analysis	Breast cancer/BC all-cause mortality	0.964 (0.761–1.221)/ 0.652 (0.488–0.873)
Kuo et al. [47]	2019	Cohort	Prostate cancer	0.69 (0.49–0.96)
Hoiso et al. [48]	2019	Cohort	Breast cancer	0.97 (0.89–1.05)
Xiao et al. [49]	2020	Meta-analysis	Lung cancer/survival	0.78 (0.70–0.86)/0.65 (0.55–0.77)
Zhang et al. [50]	2021	Meta-analysis	Any cancer	0.70 (0.65–0.76)
Kim et al. [51]	2022	Cohort	Pancreatic cancer	1.116 (0.648–1.923) ^e^ & 2.769 (1.003–7.642) ^f^
Hu et al. [52]	2023	Nested case–control	Pancreatic cancer	0.82 (0.69–0.98)
Orchard et al. [53]	2023	Cohort	Any cancer/cancer mortality	0.68 (0.51–0.90)/ 0.72 (0.43–1.19)

^a^: Relative risk is used as a generic term that includes rate ratio, hazard ratio, and odds ratio; ^b^: for more than 3 years use; ^c^: metformin use for 5 years; ^d^: metformin use for 5–10 years; ^e^: in male patients; ^f^: in female patients.

**Table 2 ijms-25-04083-t002:** Metformin impact on the IC50s of chemotherapeutic agents in in vitro trials using MTT assay.

Chemotherapy	Cancer Type	Cancer Cell Line	IC50 of Drug Alone	IC50 of the Drug + Metformin	Metformin Dose	Reference
Cisplatin	Oral squamous cell carcinoma	HSC3	17.44 ± 1.10 µM	8.32 ± 0.92 µM	10 μM	[54]
		SCC3	9.86 ± 1.55 µM	3.59 ± 1.02 µM	10 μM	
		TCA8113	9.83 ± 1.30 µM	5.73 ± 0.77 µM	10 μM	
	Ovarian cancer	SKOV3	14.35 µg/mL	11.20 µg/mL	10 mmol/L	[55]
		SKOV3/DPP	70.26 µg/mL	6.21 µg/mL		
	Lung cancer	A549	20.4 µM	15.4 µM	10 mM	[56]
Methotrexate	Ovarian cancer	SKOV3	4.21 µg/mL	2.80 µg/mL	10 mmol/L	[55]
		SKOV3/DPP	15.27 µg/mL	2.74 µg/mL	10 mmol/L	
	Hepatocellular carcinoma	HepG2	29.8 ± 0.6 nM	14.6 ± 0.8 nM	2.5 mM	[57]
		HepG2/MTX	219 ± 8 nM	17 ± 1 nM		
Doxorubicin	Breast cancer	MCF7	0.283 ± 0.036 µM	0.253 ± 0.031 µM	10 mM	[58]
		MCF7/Dox	3.23 ± 0.14 µM	1.182 ± 0.1 µM	10 mM	
Paclitaxel	Prostate cancer	PC-3 cells	13.170 ± 1.12 nM	5.423 ± 0.734 nM	5 mM	[59]
	Breast cancer	T47D cells	0.2 mg/mL	0.048 mg/mL	40 mg	[60]

**Table 3 ijms-25-04083-t003:** Completed clinical trials investigating anticancerous impact of metformin.

Metformin Monotherapy							
NCT No.	Year ᶴ	Cancer Type	No. of Patients Included	Diabetes Status	Study Design	Results	
NCT00930579 [61]	2014	BC (DCIS)	35	Non-diabetic	Phase II	No proliferation changes though reduction in relevant biomarkers was observed	
NCT01447927 [62]	2015	Barret’s Esophagus	74	Non-diabetic	Phase II	No significant change in pS6K levels	
NCT02376166 [63]	2017	Prostate cancer	14	Non-diabetic	_	Metformin was well tolerated and exhibited minimal anti-PCa activity	
NCT01266486 [64]	2018	BC	41	Non-diabetic	Phase II	There are two distinct metabolic responses to metformin: the OXPHOS transcriptional response (OTR) and FDG response. The OTR was resistant to metformin, manifested by increased proliferation. Mitochondrial response to metformin in primary breast cancer may define the anti-tumor effect.	
NCT03118128 [65]	2018	ALL	102	Non-diabetic	Phase II	Metformin + chemotherapy is effective in patients with high ABCB1 gene expression	
NCT01312467	2019	CRC	32	Non-diabetic	Phase II	Non-significant change in pS6K Ser235	
NCT01101438 [66]	2022	Early BC	3649	Non-diabetic	Phase III	In high-risk operable BC, metformin did not improve the DFS.	
**Metformin Added to Conventional Chemotherapy**							
**NCT No.**	**Year ᶴ**	**Cancer Type**	**No. of Patients Included**	**Diabetes Status**	**Study Design**	**Chemo-Drug**	**Results**
NCT01941953 [67]	2014	Refractory metastatic CRC	22	Non-diabetic	Phase II	Fluorouracil, leucovorin	Anticancer activity and better response to treatments
NCT01971034 [68]	2015	Metastatic pancreatic cancer	41	Non-diabetic	Phase II	Paclitaxel	Poor tolerance and no prognostic value in patients
NCT01210911 [69]	2015	Pancreatic cancer	121	Non-diabetic	Phase II	Gemcitabine erlotinib	No additional outcome improvement
NCT00490139 [70]	2017	Breast cancer	8381	Diabetic and non-diabetic	Phase III	Trastuzumab, lapatinib, or their combination	Improved the bad prognosis, mainly in primary HER2- and HR-positive breast cancer.
NCT01666730	2018	Metastatic pancreatic cancer	31	Diabetic and non-diabetic	Phase II	Oxaliplatin, fluorouracil, leucovorin calcium	According to RECIST, ≈50% of patients benefited clinically from metformin use
NCT01589367 [71]	2019	ER-positive breast cancer	153	Non-diabetic	Phase II	Letrozole	>10% higher response rate and more patients with Ki67 < 10%
NCT01310231 [72]	2019	Metastatic breast cancer (MBC)	40	Non-diabetic	Phase II	Anthracycline, platinum, taxane, capecitabine/vinorelbine	No significant effect on RR, PFS, or OS
NCT02325401	2020	Locally advanced HNSCC	20	Non-diabetic	Phase I	Cisplatin and chemoradiation	High OS (≈83.33% for 2 g metformin and ≈100% for 2.5 and 3 g) and 100% PFS with all metformin doses
NCT02048384 [73]	2020	Pancreatic adenocarcinoma	22	N/R	Phase Ib	Rapamycin	Well tolerated, and stable disease associated with exceptionally long survival was achieved
NCT01796028 [74]	2021	Prostate neoplasms	100	Non-diabetic	Phase II	Docetaxel	Failed to improve the outcome
NCT04143282 [75]	2021	MBC	50	Non-diabetic	Phase II	Gemcitabine + carboplatin/paclitaxel; FAC; AC; vinorelbine; capecitabine; paclitaxel	Improved radiologic RR and, better yet, insignificant OS and PFS
NCT02115464 [76]	2021	LA-NSCLC	54	Non-diabetics	RCT: Phase II	Cisplatin ± radiotherapy	Worse treatment efficacy and more toxic effects
NCT02755844	2022	Recurrent endometrial cancer	35	Diabetic and non-diabetic	Phase I/II	Cyclophosphamide and olaparib	Significant non-progression rate in recurrent advanced or metastatic endometrial cancer
NCT05351021 [77]	2023	Breast cancer	73	Non-diabetic	Phase II	Paclitaxel	Remarkable protection against paclitaxel-induced PN
NCT02949700 [78]	2023	Head and neck cancer	16	Non-diabetic	Phase I/II	Cisplatin-based chemoradiation	2-year PFS = 90% and OS = 85%. Yet, the small sample size renders effectiveness of metformin as chemo-radiosensitizer unclear
NCT04170465 [79]	2023	Primary breast cancer	70	Non-diabetic	RCT: Phase II	AC-T	Better control of chemotherapy-induced toxicities
NCT05840068 [80]	2023	MBC	107	Non-diabetic	Phase II	N/R	No significant IGF-I reduction in MBC patients on metformin
NCT03243851 [81]	2023	Glioblastoma	81	Non-diabetic	Phase II	Temozolomide	Well tolerated, but no clinical benefit in recurrent/refractory GBM

ᶴ Year: Year of the trial’s publication, or year of final completion if there is no publication available; ALL: acute lymphoid leukemia; BC: breast cancer; CRC: colorectal cancer; DCIS: ductal carcinoma in situ; HR: hormonal receptor; LA-NSCLC: locally advanced non-small-cell lung cancer; MBC: metastatic breast cancer; N/R: not reported; DFS: disease-free survival; OS: overall survival; PFS: progression-free survival; RR: recurrence rate.

## 3. Mechanism of Action of Anti-Tumorigenic Effect of Metformin

Metformin exerts its influence on multiple facets of cancer cell biology, encompassing energy levels [82], metabolism [83], cellular growth and proliferation, angiogenesis [84], and programmed cell death. The impact of metformin on cancer cells involves the indirect modulation of insulin-dependent pathways as well as direct effects through insulin-independent pathways (Figure 1). Importantly, these pathways exhibit interplay and mutual interaction.

### 3.1. Metformin Direct Effect (Insulin-Independent)

Metformin exerts its influence on cancerous cell metabolism through the inhibition of respiratory complex I, also known as NADH-coenzyme Q oxidoreductase, which is a component of the electron transport chain (ETC) located in the mitochondria (Figure 2). The inhibition of complex I leads to a decrease in the flow of electrons to complex III, where ROS are generated [12]. As a result, the production of ROS, oxidative stress, and DNA damage are reduced, thereby lowering the risk of mutagenesis. Furthermore, complex I inhibition leads to mitochondrial dysfunction and cellular energy stress, resulting in a depletion of adenosine triphosphate (ATP) and an increase in the ratio of AMP to ATP [85,86,87]. It is worth noting that the activation of AMP-activated protein kinase (AMPK) is a significant consequence of increased AMP levels, which can occur through various pathways. These pathways include direct allosteric modulation, the LKB1-mediated phosphorylation of the α catalytic subunit at Thr172, and the inhibition of Thr172 dephosphorylation by AMPK phosphatases [88]. AMPK is a serine/threonine protein kinase composed of three subunits that plays a crucial role in regulating cellular energy metabolism [87]. Besides AMP, upstream kinases, such as liver kinase B1 (LKB1) [89], Ca^2+^/calmodulin-dependent protein kinase kinase (CaMKK) [90], and TGFβ-activated kinase-1 (TAK1) [89], can activate AMPK. Once activated, AMPK maintains energy homeostasis by inhibiting anabolic processes and promoting catabolic pathways [91] (Figure 2).

AMPK exerts its effects primarily by inhibiting the growth regulator mammalian target of rapamycin (mTOR) through phosphorylation or degradation [92,93,94,95]. Phosphorylation-mediated inhibition targets either the raptor subunit of mTOR or the tuberous sclerosis complex 2 (TSC2), while mTOR degradation is stimulated by unc-51 like kinase 1 (ULK1), a crucial regulator of autophagy [95]. Interestingly, metformin can inhibit mTOR independently of AMPK by inhibiting the ragulatory complex, inactivating the RAG GTPases, and dissociating mTORC1 from its activator RHEB [96]. It can also enhance the expression of the mTOR negative regulator, DNA-damage-inducible transcript 4 protein (DDIT4, REDD1) [97]. Ultimately, mTOR deactivation inhibits key proteins involved in mRNA translation, such as ribosome S6 protein kinase (p70S6K) and eIF4E-binding proteins (4E-BP1), thereby impeding cancer cell proliferation. Furthermore, mTOR inhibition disrupts the activity of hypoxic inducible factor (HIF-1α), a crucial transcriptional regulator that aids cells in adapting to hypoxia and contributes to cancer cell resistance in radiotherapy [3]. Metformin-mediated AMPK activation also dephosphorylates insulin receptor substrate-1 (IRS-1), impeding signal transmission from the insulin receptor (IR) and insulin-like growth factor (IGF-1R) receptor to growth-promoting pathways, such as the phosphatidylinositol 3-kinase/Protein kinase B (PI3K/AKT) pathway [98]. This cascade also negatively affects mTOR signaling; however, several regulatory pathways counteract this impact on mTOR [99,100].

Various studies have demonstrated the impact of metformin-induced AMPK activation on the proliferation of cancer cells. One such mechanism involves the induction of the DICER1 gene by metformin. DICER1 encodes the DICER enzyme, which belongs to the RNase III family and is responsible for processing miRNA molecules, thereby influencing gene expression patterns [101,102]. It is worth noting that the downregulation of DICER has been associated with poor prognoses in various types of cancer [103,104,105]. Consequently, metformin affects the expressions of multiple miRNAs, which, in turn, modulate target genes involved in metabolic and oncogenic pathways. These miRNAs include miR-21, miR-26a, miR-33a, miR-140-5p, miR-142-3p, miR-181a, miR-192, miR-193b, R-20mi0, miR-205, miR-222, let-7a, and let-7c [106,107]. Notably, by manipulating Let-7, metformin inhibits the proto-oncogene c-MYC, which is known to be overexpressed in many cancers and plays a critical role in growth control, differentiation, and apoptosis [108,109]. Furthermore, metformin interacts with the tumor suppressor protein p53, although the nature of their relationship has been a subject of controversy. Upon metformin-mediated AMPK activation, p53 induction has been observed, leading to the subsequent inhibition of the AKT and mTOR pathways and resulting in cell cycle arrest [110]. Additionally, it has been observed that p53 forms a complex with LKB1, known as the LKB1-p53 complex, which ultimately leads to AMPK activation [111].

Moreover, metformin has been shown to induce the phosphorylation of programmed death ligand 1 (PD-L1) in an AMPK-dependent manner. This phosphorylation event triggers the degradation of PD-L1 and subsequently promotes the T-lymphocyte-mediated cell death of tumor cells [112,113,114]. Apart from PD-L1, metformin has also been found to interact with other immune regulators, including inhibitory immune checkpoints, M2-like tumor-associated macrophages, regulatory T cells, and myeloid-derived suppressor cells (MDSCs), thereby inhibiting immune destruction.

### 3.2. Metformin Indirect Effect (Insulin-Dependent)

Metformin indirectly affects cancer cells by decreasing the glucose and insulin levels in the body. It achieves this by targeting three key organs: the liver, muscles, and intestines. In hepatocytes, metformin stimulates IRS-2 activity and enhances the insulin-mediated suppression of gluconeogenesis. It also promotes glucose uptake by translocating glucose receptors (GLUT-1) to the plasma membrane [115,116]. Additionally, metformin inhibits the respiratory complex I of the mitochondria in hepatocytes, leading to a decrease in ATP production. This reduction in ATP hampers hepatic gluconeogenic flux, as gluconeogenesis relies on ATP [117]. The increase in the hepatic AMP levels inhibits adenylate cyclase, thereby downregulating cAMP-Protein kinase A (cAMP-PKA) activity. This inhibition further suppresses gluconeogenesis by reducing the activity of gluconeogenic flux enzymes and inhibiting the CREB-1 transcription factor, which controls the expressions of gluconeogenic genes [118,119]. Moreover, metformin suppresses inositol 1,4,5-triphosphate receptors (I3PRs), which, in turn, inhibits CREB-regulated transcription coactivator 2 (CRTC2). CRTC2 interacts with CREB-1 to activate the expressions of gluconeogenic genes [98,120]. Metformin also counteracts the gluconeogenic activity of glucagon [116,118]. Furthermore, metformin activates AMPK in skeletal muscles, leading to increased glucose uptake. This is achieved by upregulating the expression of IRs and facilitating the translocation of glucose receptors (GLUT-4) to the plasma membrane [121]. Additionally, metformin-mediated AMPK activation suppresses fat metabolism, which contributes to the reduced expressions of gluconeogenic genes [122].

However, within the intestines, metformin functions as a facilitator for the actions of glucagon-like peptide 1 (GLP-1) by augmenting the expression of GLP-1R and elevating the GLP-1 plasma levels [123,124,125]. When faced with glucose, GLP-1 heightens insulin secretion, diminishes glucagon, and exerts various metabolic effects specific to different tissues, ultimately reducing the glucose levels [124]. Furthermore, numerous in vivo trials have demonstrated that metformin reduces the levels of DDP-4, which is responsible for the degradation of GLP-1 [126,127,128,129]. However, in vitro experiments have failed to establish a direct inhibitory effect of metformin on DDP-4 [125,130]. Instead, it is now hypothesized that metformin directly enhances GLP-1 production by increasing the expressions of its precursor proteins in the large intestine, such as pre-proglucagon and proglucagon, through a β-catenin/TCF7L2-mediated mechanism [131,132,133]. Indirectly, metformin inhibits the farnesoid X receptor (FXR) via an AMPK-mediated mechanism, leading to an increase in the bile acid pool, which subsequently stimulates the TGR5 bile acid receptors on the L cell. This stimulation triggers GLP-1 secretion through mitochondrial oxidative phosphorylation and calcium influx [134,135]. Additionally, metformin’s ability to modify the composition of intestinal microbes, such as by increasing *Akkermansia muciniphila* [136], altering enterocyte glucose metabolism, and delaying gastric emptying, contributes to its hypoglycemic effect [137].

The anti-tumorigenic effect of metformin is attributed to its ability to reduce glucose and insulin levels [1,138]. In various cancer types, hyperinsulinemia, which is associated with obesity, insulin resistance, and T2DM, promotes tumorigenesis through the activation of IR-A isoforms [1,138]. Additionally, the IGF-IR has been implicated in several cancer types, either on its own or through hybridization with IR-A. Both receptors play a role in growth-promoting pathways, such as the PI3K/AKT/mTOR and RAS/RAF/MAPK signaling network [139]. By reducing the concentration of insulin and IGF-1, metformin not only decreases the downstream signaling pathways of these receptors in cancer cells but also downregulates other molecules that promote tumor growth, including growth factors, sex hormones, proinflammatory cells, cytokines, and metabolic intermediates. Furthermore, metformin prevents the unfolded protein response (UPR) in an AMPK-dependent manner, which is crucial for cell survival under stress conditions, such as low glucose levels [3,140]. It achieves this by activating the PERK/ATF4/CHOP axis and inhibiting the ATF6/GRP78 axis [141]. Consequently, metformin induces apoptosis in cancer cells [142]. In addition to its hypoglycemic effect, metformin also influences the inflammatory process involved in tumorigenesis. Studies have shown that metformin specifically inhibits the nuclear translocation of NF-κB and the phosphorylation of STAT3, thereby suppressing the inflammatory response associated with the growth of cancer stem cells (CSCs) and cellular transformation [143,144]. Noteworthy, CSCs are associated with drug resistance and tumor relapse [145]

## 4. Metformin and Cancer Prevention

Metformin not only affects the microenvironment of cancer cells but also influences various factors that contribute to tumorigenesis, such as diabetes, cancer-promoting alterations associated with aging, hyperlipidemia, and obesity.

### 4.1. Diabetes

In a meta-analysis conducted by Vigneri et al., it was found that patients with diabetes had higher incidences of many types of cancer. Specifically, diabetes was associated with a 2-fold higher risk of liver, pancreatic, and endometrial cancers, and a 1.2–1.5-fold higher risk of colon, rectum, breast, and bladder cancers [146]. Diabetic patients on metformin not only exhibited a reduced incidence of cancer but also demonstrated lower all-cause mortality [147,148]. For instance, there was an observed increase in the complete-response rates among breast cancer patients treated with metformin [149]. Additionally, another study showed that diabetic patients diagnosed with prostate cancer experienced a significant decrease in the risk of cancer-specific and all-cause mortality with prolonged metformin treatment [150]. Furthermore, research by Tan et al. indicated that diabetic patients with advanced lung cancer had improved overall survival and longer progression-free survival with metformin treatment [151]. Remarkably, metformin has demonstrated efficacy in impeding the advancement of precancerous conditions to cancer even in non-diabetic individuals. For instance, in a preclinical study, metformin displayed a notable capacity to restrain oral tumor lesions and halt the progression of precancerous lesions to squamous cell carcinomas [152]. Similarly, the protective effect of metformin in preventing adenoma recurrence in colorectal patients was demonstrated [153]. Moreover, studies have indicated that metformin can induce regression in precancerous states [154,155].

### 4.2. Aging

When it comes to cancer, aging is considered one of the primary risk factors [156]. Metformin has shown promising effects on mechanisms related to aging and the mitigation of age-related ailments, including cancer. Numerous studies have linked the use of metformin to a reduced occurrence of cancer [154], as well as to its ability to improve aging-associated characteristics by enhancing nutrient sensing, promoting autophagy, and protecting against macromolecular damage [157] (Table 4). Interestingly, long-term use of metformin in diabetic patients has been found to enhance survival in various types of cancer [155,158,159]. Importantly, the impact of metformin on aging-related diseases has been shown to be independent of its anti-diabetic effect [160]. These findings confirm the role of metformin in protecting against aging, although the exact mechanism is still unclear. However, metformin has been shown to influence a range of aging processes. One such mechanism is the activation of AMPK, which regulates cellular metabolism and energy, thereby promoting healthy aging. Metformin enhances cellular resilience and metabolic function and counteracts age-related degradation by modulating AMPK/mTOR [161]. Another proposed mechanism for metformin’s anti-aging effect is its ability to reduce oxidative damage to cells by suppressing ROS production from the mitochondrion ETC, as described earlier.

Indeed, metformin acting on mitochondria produces an anti-aging effect as well. This is because mitochondrial dysfunction contributes significantly to aging, as it impairs cellular metabolism and homeostasis. Karnewar et al. [162] recently reported that metformin directly stimulates SIRT1, leading to an upregulation of DOT1L and an increase in the trimethylation of H3K79 (H3K79me3). This, in turn, results in an increase in SIRT3, a major mitochondrial biogenetic marker. The upregulation of SIRT3 not only promotes mitochondrial biogenesis but also delays the senescence process by increasing PGC-1α.

Senescence is an irreversible cell cycle arrest process that has been strongly implicated in aging and aging-related diseases, including cancer. Over time, senescent cells secrete a senescence-associated secretory phenotype (SASP) and accumulate, creating an inflammatory microenvironment that promotes aging-related diseases [163]. Various studies have shown that metformin inhibits SASP and cellular senescence through different mechanisms. These mechanisms include the regulation of MBNL-1/miR-130a-3p/STAT3 [164], the upregulation of SIRT1 [165], the downregulation of NLRC4 [166], and the downregulation of NF-κB signaling [167]. Although the connection between these anti-senescence effects and the anticancer effect of metformin has yet to be explored, different cancers have been reported to exhibit SIRT upregulation [168], p53 downregulation [169], and mTOR attenuation [170]. It is important to note, however, that the modulatory effect of metformin on senescence should be interpreted cautiously, as it has been reported to increase SASP and senescence in various cancer cells and to even help cancer cells evade senescence [171,172,173]. This ambiguity in the current knowledge highlights the need for further investigations in this area.

Furthermore, the expressions of antioxidant genes are enhanced by metformin through the SKN-1/Nrf2 transcription pathway [174]. The activation of this pathway leads to the activation of several antioxidant enzymes, including superoxide dismutase and catalase, which effectively neutralize ROS and safeguard cells against oxidative stress. In addition, another significant mechanism involved in the regulation of aging is the targeting of stem cells. Over time, the regenerative capacity of stem cells diminishes, contributing to age-related diseases. Fang et al. discovered that a low dose of metformin increases the expression of GPx7, a glutathione peroxidase localized in the endoplasmic reticulum, through the involvement of nuclear factor erythroid 2-related factor 2 (Nrf2). This upregulation of GPx7 helps delay premature cellular senescence [175]. It is worth noting, however, that metformin exhibits a cytotoxic effect on the expressions of CSCs, as evidenced by the inhibition of various CSC markers.

### 4.3. Hyperlipidemia and Dyslipidemia

Hyperlipidemia and hypercholesterolemia are modifiable factors that have been identified as significant contributors to the development of cancer [176]. In fact, dyslipidemia has been found to have a detrimental impact on the clinical outcomes in cancer patients, as it facilitates tumor metastasis [177], promotes chemoresistance [178], and increases the cytotoxicity of chemotherapies [179]. However, it is important to note that the findings regarding the association between lipid profiles and different types of cancer have been conflicting. For example, studies have shown that high levels of cholesterol and reduced levels of HDL are positively correlated with the aggressiveness of breast cancer [176,180]. Similarly, elevated total cholesterol levels have been closely linked to prostate cancer and testicular cancer, while high cholesterol levels have been associated with a lower risk of liver cancer, stomach cancer, and lymphatic cancers [176,181]. In a cross-sectional study conducted by Ghahremanfard et al., it was observed that the triglyceride levels were elevated in ovarian cancer but reduced in colorectal cancer (CRC) [182]. Additionally, the same study found that the total cholesterol and LDL levels were elevated in breast cancer but diminished in gastric cancer (GC).

Cholesterol plays a role in the development of tumors through various mechanisms. When it directly binds to the G-protein-coupled receptor, it activates the oncogenic Hedgehog signaling pathway, which is associated with cell differentiation, cell proliferation, and the formation of cancer [183,184]. Cholesterol has also been found to bind to the PDZ domain of NHERF1/EBP50 [185], a key contributor to the PI3K/Akt and Wnt/B-catenin pathways, thereby promoting the development of cancer [186]. Additionally, lysosomal cholesterol stimulates mTOR through the SLC38A9-Niemann-Pick C1 signaling complex [187]. In addition to cholesterol itself, its anabolic and catabolic metabolites also play a role in cancer metastasis and overall aggressiveness [186]. Two particularly notable metabolites are mevalonic acid (MVA) and isoprenoids. MVA serves as a precursor to cholesterol and activates mTOR and NF-κB, leading to changes in apoptosis and the cell cycle. Interestingly, MVA is closely linked to isoprenoids in terms of cancer progression. As a first-line therapy, statins inhibit the cholesterol pathway by blocking HMG-CoA reductase, the enzyme that is the rate-limiting step in cholesterol synthesis [188]. This alteration in the cholesterol pathway also affects the production of MVA and isoprenoids, as they are derivatives of HMG-CoA [189]. Furthermore, metformin has been found to impact cancer by modulating pathways specific to cholesterol synthesis. When combined with statins, metformin has been shown to result in a more than 50% reduction in cancer mortality in patients with high-grade prostate cancer, compared to those treated with statins alone [190].

Several studies have been conducted to investigate the specific mechanisms underlying the cholesterol-lowering effects of metformin. A study by Hu et al. revealed that the introduction of metformin to mouse cells led to an increase in glycolysis, resulting in elevated lactate dehydrogenase (LDH) activity and a reduction in the translocation and formation of PCSK9 [191]. PCSK9 plays a crucial role in cholesterol homeostasis, and its decrease leads to an upregulation of LDL receptors, ultimately causing a decrease in the free cholesterol levels. Additionally, a study by Sharma et al. demonstrated a significant decrease in the expressions of cholesterol regulatory genes (such as HMG-CoA reductase and SREBP1) upon the introduction of metformin to breast cancer cells, which are known for their high cholesterol content [192]. The impact of metformin on cholesterol levels extends to its effects on cancer metastasis, the epithelial–mesenchymal transition (EMT), and stemness, as observed in diffuse large B-cell lymphoma (DLBCL), a high-grade non-Hodgkin’s lymphoma [193]. The most prominent pathway identified in this study, which involved three DLBCL cell lines, was the B-cell receptor (BCR) signaling pathway and the biosynthesis of cholesterol. The administration of metformin inhibited the growth of the lymphoma by targeting HMGCS1, phosphorylated SYK, and AKT, which are typically activated by BCR signaling [193]. This attenuation of downstream signaling ultimately leads to a decrease in the cholesterol levels in the body, thereby halting the survival and colonization of the lymphoma [193].

### 4.4. Obesity

Obesity is a prevalent condition that increases the susceptibility to numerous chronic illnesses, including T2DM, cardiovascular disease, and, notably, cancer [194]. It has been documented that obesity is not only linked to a higher risk of various cancer types but also to elevated rates of cancer recurrence and mortality [195,196]. However, the molecular mechanisms connecting obesity and cancer remain incompletely comprehended. In the context of obesity, dysregulated fatty acid secretion and metabolism, anabolic and sex hormone secretion, immune dysregulation, and chronic inflammation contribute to the development and spread of cancer [197,198].

Several trials have reported the ability of metformin to reduce weight gain and promote weight loss in diabetic and non-diabetic obese subjects [199,200,201]. It is believed that metformin affects weight by influencing adipose tissues. Previous research conducted by Kim et al. [202] demonstrated that metformin induces weight loss by modulating fibroblast growth factor 21 (FGF21), a hormone that enhances lipolysis in white adipose tissue and prevents fat accumulation. Subsequent preclinical studies have revealed that metformin enhances the metabolic activity of brown adipose tissue, thereby preventing weight gain [203,204,205]. In brown adipose tissues, the metformin-induced activation of AMPK and FGF21, along with its ability to regulate various thermogenic markers, such as UCP1, NRF1, and PGC1α, contribute to its weight loss effects [206]. Metformin has also been shown to exert control over appetite. The suppression of appetite with metformin has been attributed to lactate-mediated mild metabolic acidosis [207,208,209], the increased production of GLP-1 mediated by the bile acid pool [210], and increased expressions of leptin receptors in the hypothalamus [211]. Interestingly, the potential contribution of metformin’s effect on obesity to its anticancer impact has yet to be established. However, due to its ability to lower glucose, insulin, and free fatty acids, as well as to disrupt the insulin/IGF-1-PI3K/AKT/mTOR and fatty acid/lipid biosynthetic pathways, metformin may serve as a metabolically targeted therapy for obesity-driven cancers, thereby eliminating the association between obesity and cancer [109,212,213,214]. Furthermore, recent studies investigating the impact of visceral obesity, rather than a high BMI, in NSCLC and CRC have shown that metformin mitigates the resulting poor prognostics [215,216].

## 5. Metformin Use in Different Cancer Types

### 5.1. Breast Cancer

Breast cancer (BC) is a prevalent form of cancer worldwide, with a significant number of newly diagnosed cases and death cases in 2020 [217]. The current treatment for BC involves various approaches, such as surgery, radiotherapy, chemotherapy, endocrine therapy, and targeted therapy [218]. However, the development of chemoresistance and metastasis poses challenges in the treatment of BC, leading to a poor prognosis. Metformin has attracted considerable attention due to its potential impact on breast cancer. Previous studies have indicated that the long-term use of metformin can reduce the risk of breast cancer in diabetic women [24]. Additionally, it has been reported that long-term metformin treatment can decrease the risk of ER-positive breast cancer [219]. A recent meta-analysis involving a large number of BC patients demonstrated that metformin not only increased the complete/partial response rate in BC but also suppressed various BC biomarkers, including HOMA-IR, insulin, sex hormones, SHBG, Ki67, obesity, hs-CRP, caspase-3, p-Akt, blood glucose, and the lipid profile [220]. Furthermore, the addition of metformin to neoadjuvant chemotherapy and ERBBS-targeted therapy improved the pathological complete response (pCR) in HER2-positive BC patients with a specific genetic variant [221].

Metformin has been found to have various effects on breast cancer. In a study by Vazquez-Martin et al., it was discovered that metformin inhibits the growth of breast cancer cells by reducing the levels of HER2 through the modulation of the AMPK/mTOR/p70S6K1 axis [222]. Another study by Hu et al. reported that low-dose metformin in the BT-549 cell line suppressed the expressions of stemness markers such as CD44, Nanog, OCT-4, and c-mym, leading to cell cycle arrest by increasing FOXO3 and p53 [223]. Metformin was also found to reduce the serum levels of estradiol in patients, which may be a possible mechanism for its ability to resist breast cancer development [224]. Furthermore, metformin inhibited the expression of cyclooxygenase (COX) 2, which is known to promote breast cancer proliferation and angiogenesis, thereby limiting the metastasis of breast cancer [225,226]. Additionally, several studies have reported the regulatory effects of metformin on miRNAs, as shown in Table 5 [227]. Through an AMPK-dependent mechanism, metformin has been shown to decrease the levels of HIF-1α at both the mRNA and protein levels, thereby exhibiting antiproliferative and anti-Warburg potential in breast cancer [110]. Metformin also inhibits angiogenesis by targeting the HER2/HIF-1α/VEGF secretion axis [228].

In addition to inhibiting the growth of cancer cells, a study by Zimmermann et al. revealed that metformin demonstrated a synergistic effect when combined with fulvestrant, an estrogen receptor antagonist, in ER-positive breast cancer cells [234]. This combination led to cell cycle arrest at the G1 phase by enhancing the expression of Cyclin G2 and the cell cycle arrest induced by fulvestrant. Another study found that metformin inhibited the growth of MCF7 cells by causing cell cycle arrest at the G0/G1 phase [235]. This was achieved through the regulation of the AMPK/mTOR/cyclin D1 axis, preventing the cells from entering the S phase. Furthermore, the concurrent administration of metformin with paclitaxel induced robust cell cycle arrest at the G2/M phase by modulating the AMPK/mTOR signaling pathway [236]. Additionally, a recent study demonstrated that metformin caused the regression of MDA-MB231, a metastatic breast cancer cell line, and sensitized it to cyclophosphamide by downregulating PDGF-B and normalizing the vessels [237].

Remarkably though, several studies had contradictory results, making the anticancerous role of metformin in BC questionable. In a recent RCT trial of 3649 by Goodwin et al. [66], it was reported that there was a non-significant impact of metformin on the invasive disease-free survival in non-diabetic patients with high-risk operable BC. Furthermore, the use of metformin was found to be associated with an increased risk of ER-positive BC according to Park et al. [219]. Factually, in BC, the response to metformin is influenced by various factors. As indicated by further studies, the protective role of metformin is dependent on the type of BC, the hormonal levels, and the glucose level. For example, Zhouang et al. [236] described that the effects of metformin-mediated AMPK activation in breast cancer cells varied depending on the specific cell line. In fact, metformin treatment resulted in growth arrest in five breast cancer cell lines (MCF7, BT20, T47D, MDA-MB-453, and MDA-MB-474), while MDA-MB-231 showed resistance to the effects of metformin. The response of breast cancer to the anticancerous effect of metformin also varies based on the availability of glucose. High glucose levels not only promote breast cancer regression but also diminish the antiproliferative and pro-apoptotic effects of metformin [238,239,240]. Conversely, under normoglycemia-like conditions, metformin-mediated lethality in breast cancer was observed in vitro [241].

### 5.2. Colorectal Cancer (CRC)

CRC is the third most prevalent cancer and the second deadliest worldwide [242]. Though both obstructing and non-obstructing colon tumors have the best tumor prognosis if resected surgically [243], non-resectable ones must be treated by chemotherapeutic agents, like oxaliplatin and 5-fluorouracil (5-FU) [243]. When considered as an adjuvant therapy in CRC, metformin has been proven effective, as it decreases the CRC proliferation, stemness, and metastatic activity [244]. A study concerning aberrant crypt foci (ACF)-positive mice demonstrated that, with metformin, the number was significantly lowered, and S6 kinase, p-mTOR, and S6 protein were decreased [245]. Similarly, when studied in KRAS-mutated human cells, which are resistant to chemo-drugs, metformin caused CRC cell cycle arrest at the G1/S phase [246]. Cyclin D1, a CDK protein, was inhibited via the regulation of its activation pathway: RAS/RAF proto-oncogene signaling [246].

Metformin-induced apoptosis has been linked to Bcl-2, Bax, Caspase-3, Mcl-1, and TRAIL [247]. The initial three molecules were linked to an increase in tumor-infiltrating lymphocytes as well as a decrease in CD163 (+) M2 cells caused by metformin [248]. Metformin has been linked to increased TRAIL-induced apoptosis in CRC and Mcl-1 reduction by ubiquitination and degradation [249]. Recently, a study by Xiao et al. identified the metformin-mediated inhibition of Inhibin βA, halting CRC cell proliferation by hindering the TGF-β/PI3K/Akt signal transductions [250]. Moreover, a significant positive correlation suggests that metformin may interfere with the EMT process and improve the CRC prognosis [251]. Metformin has also been shown to affect HT29 and p53^−/−^ CRC cells by targeting their stem cells. A noted decrease in the expression of the cyclin D1 CDK and c-Myc protein suggested cell cycle arrest, which was supported by an increased percentage of cells arrested at the G0/G1 phase [252].

The effect of metformin is prominent when combined with anti-CRC agents. For instance, the numbers of tumor cells in mice were remarkably reduced when exposed to metformin and fluorouracil/oxaliplatin compared to no metformin use [253]. 5-FU and metformin both work synergistically to alter the NF-κB pathway to reduce gastric inflammation [254]. When the two drugs were tested against SNU-C5/5FuR, a 5-FU-resistant CRC cell line, there was a decreased expression of NF-κB, the factor responsible for cytokine-mediated inflammation, through AMPK/mTOR signaling [254]. Metformin also enhances the actions of cisplatin against CRC cells by activating the PI3/Akt pathways [255]. Furthermore, metformin boosted the cyclophosphamide efficacy in non-angiogenic CT-26 cells, which are chemoresistant, by enhancing vascularization and blocking caspase-mediated endothelial apoptosis [256].

Perhaps the most crucial and specific pathways in CRC are the Wnt/β-catenin signaling pathways. These pathways initiate the activation of the destruction complex, which is typically degraded by the proteasomes in the vicinity [257]. When mutated, the β-catenin, a component of the destruction complex, evades the proteasome and stimulates tumor cell proliferation [257,258]. Although there are currently no approved therapeutic interventions targeting this pathway, metformin has been hypothesized to reduce the proliferation and stemness induced by this pathway. Experiments with metformin have demonstrated a decrease in β-catenin levels, resulting in a reduction in the EMT [259]. However, there is currently insufficient evidence to directly link metformin to the Wnt/B-catenin pathway [259].

### 5.3. Gastric Cancer (GC)

Despite the advancements in medical technology and radiotherapy, the 5-year survival rate for metastatic GC remains very low. While gastrectomy is considered the main curative therapy, patients with unresectable or metastatic disease typically undergo chemotherapy. The standard pharmacological treatment involves a combination of epirubucin, cisplatin, and 5-fluorouracil (ECF), but recent studies have shown that this regimen may do more harm than good [260]. Furthermore, the survival rates for different age groups affected by gastric cancer are alarmingly low, with less than 40% surviving beyond 5 years [261]. In contrast, the use of metformin has demonstrated significant positive effects in the management of gastric carcinoma across multiple clinical trials [262,263]. A cohort study specifically examined the effects of metformin, insulin, and sulfonylureas on the risk of developing GC and found that the latter two drugs actually increased the risk of GC by elevating the levels of IGF-1, which promotes cancer cell survival [264]. However, when evaluating the prognostic impact of metformin in diabetic patients with gastric adenocarcinoma, Zheng et al. reported an improved prognosis with the use of metformin [265]. Similarly, a recent meta-analysis demonstrated that metformin use in diabetic patients with GC was associated with better overall survival and recurrence-free survival and a reduced recurrence rate following gastrectomy [266]. These findings highlight the potential of metformin as a promising therapeutic option for gastric cancer patients, particularly those with diabetes. Further research and clinical trials are warranted to fully explore the benefits and mechanisms of metformin in the treatment of GC.

Metformin has also demonstrated significant efficacy in inhibiting the proliferation of gastric cells. The activation of AMPK leads to the inactivation of acetyl-CoA carboxylase (ACC) through phosphorylation, resulting in a reduction in the proliferation of GC cells. In its dormant state, ACC prevents fatty acid synthesis, creating a state of starvation for the cancerous cells. Studies conducted on extracted human gastric cells have examined the ACC/pACC ratio and revealed a strong positive correlation between increased levels of phosphorylated ACC (pACC) and the prognosis of the cancer [267]. Another factor that contributes to GC proliferation is hepatocyte nuclear factor α (HNF-α). HNF-α promotes the cell cycle, downregulates cyclins, and facilitates the proliferation of neoplastic cells. Metformin, through AMPK signaling, inhibits HNF-α and its downstream signaling mechanisms, suggesting its potential as a diagnostic marker for future therapeutic interventions [268]. AMPK has also been demonstrated to block cell cycle progression in GC cells at the G0/G1 phase by reducing the expressions of epidermal growth factor receptor (EGFR) and insulin-like growth receptor-1 (IGF-1R), dephosphorylating retinoblastoma (RB), and inhibiting cyclin D1, CDK4, and CDK6 [269]. In this study, three GC cell lineages were identified that were inhibited through cyclin D inhibition, namely, MKN1, MKN45, and MKN74 [269].

In addition, metformin has been found to induce apoptosis in patients with gastric cancer (GC). This process is mediated by AMPK, which inhibits mTOR and mitochondrial complexes, ultimately leading to the apoptotic death of GC cells while sparing normal-functioning cells [262]. When metformin was introduced to AGS cells, the levels of p-mTOR and p-AKT, which are apoptosis regulators, were significantly reduced [270]. Interestingly, metformin has also been shown to decrease the levels of mitochondrial-dependent proteins involved in apoptosis, such as cytochrome C, phosphorylated Bcl-2, and BAD. The combination of metformin and cisplatin has been studied in relation to inducing apoptosis in GC cells, although the specific outcome and underlying mechanism of this combination remain unclear. It has been observed that the combination of metformin and cisplatin leads to a decrease in the cancer circumference and a slowdown in metastasis [271]. However, contrary to expectations, the levels of p-mTOR and p-4EBP1 were found to be significantly increased rather than decreased [271]. The antiproliferative effects of this drug combination may be attributed to their synergistic cytotoxic side effects. Furthermore, metformin has also been studied in combination with oxaliplatin, a drug closely related to cisplatin. The potencies of these drugs were found to be enhanced when used together. The levels of Bcl-2 and cyclin D were reduced, while the levels of Bax and caspase-3 were increased, indicating a pro-apoptotic environment [272].

Metformin has been found to attenuate metastasis in GC through its effects on different proteins. One such group of proteins is cadherins, which are involved in cell–cell communication. Metformin’s influence on these proteins helps prevent the migration and growth of cells in other parts of the body, a process known as the EMT [262]. The exact mechanism by which metformin inhibits the EMT is not fully understood. In addition to its effects on cadherins, metformin also inhibits metastatic peritoneal proliferation by downregulating Nf-κB, rather than by activating AMPK [262]. This was demonstrated in an experiment involving patients who had previously been treated for *H. pylori* infection and later developed gastric cancer. Metformin was shown to reduce the risk of gastric cancer, independent of the levels of HbA1c [273].

Another important aspect of cancer progression is the maintenance of cancer cell stemness. Metformin has been identified as a potential regulator of this process. One specific gene, known as sonic hedgehog (SHH), has been studied in relation to cancer and is considered a potential therapeutic target [274]. In an experiment involving GC cells, an anti-SHH preparation called cyclopamine was used. Interestingly, the number of viable GC cells decreased when exposed to cyclopamine [274]. Building on this finding, the role of metformin in regulating the SHH expression and inhibiting the CSC ability and metastatic proliferation has been investigated [274].

Despite extensive research on the interplay between GC and metformin, there are still numerous underlying mechanisms that remain unidentified. One innovative approach to comprehending the role of metformin in GC involves analyzing RNA sequences [262]. Specifically, researchers are currently investigating the impact of the metformin-induced inhibition of long noncoding RNAs (lncRNAs). Among these lncRNAs, a particular oncogenic variant called Loc 100506691 has garnered attention due to its association with GC proliferation and patient survival [275]. In GC, elevated levels of this lncRNA influence the expressions of two miRNAs, namely, miR-26a-5p and miR-330-5p. Consequently, these miRNAs inhibit the transcription of CHAC1 by targeting its 3′UTR, thereby promoting growth and metastases [275]. It is believed that metformin disrupts the Loc 100506691-miRNAs-CHAC1 axis [275]. Although the exact mechanism remains incompletely understood, this axis holds promise as a potential target for future drug interventions.

### 5.4. Liver Cancer

Primary liver cancer, the fifth most prevalent cancer worldwide, has the third highest fatality rate among all cancers. Hepatocellular carcinoma (HCC) accounts for 85% of all primary liver cancer cases. Various factors contribute to the increased risk of HCC, with hepatitis B virus (HBV) and hepatitis C virus (HCV) being particularly significant [276]. Additionally, epidemiological evidence indicates that individuals with T2DM have a from 2- to 3-fold higher relative risk of developing HCC [148,277,278]. Furthermore, in individuals without HBV or HCV infection, the coexistence of obesity and diabetes is responsible for 37% of HCC cases [279]. Research has demonstrated that the administration of metformin, a medication used to treat diabetes, is associated with a reduced risk of HCC and exerts a protective effect against its development [23,280].

In the context of inflammation-mediated tumorigenesis and the development of HCC, the NF-κB and STAT3 signaling pathways play crucial roles in regulating various downstream genes that govern cell proliferation, apoptosis, stress responses, and immune functions. Notably, the production of the STAT3-activating cytokine interleukin-6 (IL-6), which is regulated by NF-κB, has been identified as a risk factor for HCC development [281]. Therefore, targeting both pathways that control IL-6 expression and those that regulate its ability to activate STAT3 could be a promising approach for therapeutic intervention [282,283]. One potential therapeutic agent that has shown promise in modulating these pathways is metformin. Upon activation by metformin, AMPK reduces the degradation of IκBα, thereby attenuating NF-κB signaling, decreasing IL-6 expression, and inhibiting STAT3 signaling [284]. This is supported by the observation that the inhibitory effect of metformin on proliferation is significantly reduced in cells transfected with p65 (a subunit of NF-κB) or IBSR (an inhibitor of IκB degradation) [284]. Furthermore, it is worth noting that liver tumors often exhibit increased lipogenesis and fatty acid production [285]. In this regard, Bhalla et al. demonstrated that metformin can decrease HCC by inhibiting de novo lipogenesis through the suppression of key enzymes involved in this process, such as ACC, fatty acid synthase (FAS), and ATP citrate lyase (ACLY), at both the mRNA and protein levels [286]. In addition to these mechanisms, other effects of metformin in HCC have been described. These include upregulating the hippo signaling pathway by increasing the expressions of MST1, MST2, LATS1, and YAP [287], promoting KLF6/p21-mediated cell cycle arrest [288], inhibiting Shh-induced cell proliferation [289], and inhibiting anti-apoptosis proteins, such as BCL2 and MCL1 [290]. These findings highlight the multifaceted nature of metformin’s potential therapeutic effects in HCC.

Metformin has also been implicated in the prevention of liver metastasis [291]. The mechanism of metastasis in HCC is complex and involves the EMT and angiogenesis. One notable characteristic of HCC is the depletion of fatty acid transport protein-5 (FATP5), which promotes aggressive progression, the EMT, and metastasis in HCC by silencing AMPK and promoting mTOR-mediated proliferation [291]. However, the metformin-induced activation of AMPK can counteract metastasis in FATP5-deficient HCC by reversing the EMT. Furthermore, when combined with empagliflozin, metformin has been shown to decrease angiogenesis and metastasis, as evidenced by the decrease in both VEGF and MMP-2/TIMP-1 [292].

It is worth mentioning that Gao et al. demonstrated that high matrix stiffness in HCC can promote resistance to the anti-metastatic effects of metformin by upregulating integrin-β1 and its downstream pathways [293]. Apart from empagliflozin, metformin has also been found to enhance the cytotoxic effects of various HCC drugs, including aloin [294], antifolates [295], dichloroacetate (DCA) [296], celastrol [297], and sorafenib [298,299]. Additionally, metformin has been reported to sensitize HCC to sorafenib by inhibiting CXCR3 signaling, which contributes to sorafenib resistance [294]. Similarly, the combination of metformin with 5-fluorouracil has been shown to inhibit HCC proliferation, as well as the expressions of HIF-1α and multidrug resistance-associated protein 1 (MRP1) [300].

### 5.5. Lung Cancer

Lung cancer is the second most common type of cancer, accounting for 11.4% (2.21 million) of all cancer diagnoses [242]. One of the most concerning aspects of lung cancer is its high mortality rate, with 1.80 million individuals losing their lives to the disease, representing 18% of all cancer deaths [242]. Among the different types of lung cancer, non-small-cell lung cancer (NSCLC) is the most frequently diagnosed, making up approximately 80–85% of all cases [301]. When it comes to genetic mutations, NSCLC often exhibits alterations in genes such as EGFR, K-Ras, p53, and LKB1 [302]. The standard treatment for advanced stage NSCLC typically involves a combination of platinum-based chemotherapy and fractionated thoracic radiotherapy [301].

Numerous studies have highlighted the potential benefits of using metformin in the fight against lung cancer. For instance, Xiao et al. demonstrated that metformin treatment significantly reduced the incidence of NSCLC [49]. Recent research has also shown that the use of metformin is associated with a lower risk of lung cancer and improved progression-free survival in patients with advanced lung adenocarcinoma when combined with EGFR-TKI therapy [303,304]. These findings align with previous studies that have explored the impact of metformin on the lung cancer risk in diabetic patients [14,305,306]. Retrospective clinical evidence further supports these findings, indicating improved survival outcomes in patients with locally advanced NSCLC and diabetes who received metformin treatment [151,235,307].

Preclinical studies on NSCLC have demonstrated that the use of metformin activates AMPK, leading to the induction of p53, the suppression of mTOR, and the inhibition of tumor growth. This ultimately enhances the tumor’s response to radiotherapy and chemotherapy [235,308]. Remarkably, in NSCLC, metformin attenuates the PI3K/AKT and MEK/ERK signaling pathways while downregulating IGF-1R. However, in small-cell lung cancer (SCLC), it suppresses PI3K/AKT but increases MEK/ERK [309,310]. Furthermore, the addition of metformin to NSCLC A549 cells results in the upregulation of microRNA-7, which inhibits NSCLC growth and metastasis [311].

There have been reports of synergistic effects when combining metformin with anti-lung cancer treatments through various mechanisms. For example, metformin has been shown to enhance the effect of cisplatin by inhibiting the production of ROS and IL-6 secretion. This is achieved by modulating the STAT3 pathway through a mechanism independent of LKB1-AMPK [312,313,314]. Li et al. reported that metformin has the ability to reverse crizotinib resistance by inhibiting IGF-1R, thereby enhancing the cytotoxic effect of crizotinib [315]. While the role of metformin in lung cancer remains controversial, the overexpression of SIRT1 has been associated with a poor prognosis in NSCLC [316,317]. A recent study demonstrated the synergistic activity of tenovin-6 in combination with metformin, leading to the inhibition of SIRT1 expression and the suppression of cell proliferation [318].

The efficacy of metformin in lung cancer, similar to breast cancer, has been a subject of controversy due to conflicting findings. The combination of metformin and chemotherapy in non-diabetic NSCLC patients resulted in poorer treatment outcomes and increased toxic effects when compared to chemoradiotherapy alone [76]. Furthermore, the metformin group exhibited lower rates of 1-year progression-free survival and overall survival compared to the control group. Similarly, a meta-analysis conducted by Tian and colleagues found that metformin did not improve the overall survival in diabetic patients with NSCLC [319].

### 5.6. Ovarian Cancer

Metformin has demonstrated promising anticancer effects in research conducted on ovarian cancer. According to a recent cohort study, the prolonged use of metformin in ovarian cancer patients was linked to a decrease in mortality rates and improved overall survival [320]. Metformin has been shown to effectively inhibit the PI3K/AKT/mTOR signaling pathway in ovarian cancer cells, leading to cell cycle arrest at the G2/M checkpoint [321,322]. However, the impact of metformin on cancer cells seems to be less significant when glucose levels are high, whereas its cytotoxic effects are enhanced in low-glucose conditions due to the induction of ASK1-mediated mitochondrial dysfunction [323,324]. Metformin has been shown to reduce the transcription of Axl and Tyro3, two receptor tyrosine kinases associated with cell survival and resistance to apoptosis, in ovarian cancer cells [325]. It also inhibits the activation of downstream signaling molecules, including Erk and STAT3, in triple-negative breast cancers [326]. The deactivation of ERK and STAT3 can have significant consequences in cancer progression. ERK promotes tumorigenesis by inhibiting FOXO3a, leading to increased cell proliferation and tumorigenicity [327]. Similarly, STAT3 activation is associated with tumor angiogenesis and metastasis [328,329]. The inhibition of ERK-MAPK signaling can suppress angiogenesis and tumor growth [330], while targeting STAT3 can induce apoptosis and inhibit tumor cell proliferation [331]. Metformin has also been found to inhibit the growth of ovarian cancer cell lines and reduce angiogenesis, adhesion, and macrophage infiltration in both in vitro and in vivo models [321,332,333].

Metformin has been found to reduce angiogenesis in metastatic tissues, decrease the adhesion of ovarian cancer cells, and suppress the infiltration of macrophages [334]. Rattan [321] and Wu [334] both demonstrated that metformin reduces neovascularization, with Rattan attributing this to the blocking of the mTOR signaling pathway and Wu attributing it to the diminishment of the angiogenic and carcinogenic properties of platelets. Furthermore, Rattan [335] and Lengyel [322] showed that metformin suppresses tumor growth and enhances the cytotoxicity of cisplatin while also increasing the sensitivity to paclitaxel. These findings suggest that metformin could be a valuable addition to the treatment of ovarian cancer.

### 5.7. Pancreatic Cancer

Pancreatic cancer, which accounts for 2.6% of all cancer cases and ranks as the 10th most common cancer, has one of the lowest survival rates among all cancer types. In 2020, approximately 495,773 cases were reported, resulting in the deaths of an estimated 466,003 patients, making it the seventh highest cause of cancer-related deaths [242]. According to the American Cancer Society, the 5-year survival rate for all stages of pancreatic cancer is only 12% [336]. It is worth noting that pancreatic ductal adenocarcinoma (PDAC) comprises more than 90% of all pancreatic cancer cases [337].

The impact of metformin on PDAC patients remains a subject of debate. A case–control study found that diabetic patients who took metformin had a significantly lower risk of developing PDAC compared to those who did not take the medication (OR = 0.38; 95% CI, 0.22–0.69; *p* = 0.001) [338]. Similarly, a meta-analysis involving 1,535,636 patients from 37 studies reported a 46% reduced risk of pancreatic cancer among metformin users [339]. However, the exact mechanism through which metformin affects human cells, particularly in the context of PDAC, is not yet fully understood [340].

Metformin exerts its effects on PDAC through two distinct mechanisms. Firstly, it can directly impact pancreatic cells, and secondly, it can indirectly influence PDAC through systemic pathways [5,341]. In terms of its direct effects, metformin acts in both an AMPK-dependent and AMPK-independent manner to affect PDAC. KRAS mutations play a significant role in tumor initiation in PDAC [342]. The metformin-mediated activation of AMPK inhibits the IRS/PI3K/AKT pathway and mTORC1 [343,344,345], resulting in the inhibition of ERK signaling and DNA synthesis. Additionally, metformin can suppress the expression of YAP/TAZ in various cancers, including PDAC, through AMPK-mediated mechanisms [346]. Recent research has also demonstrated that metformin induces apoptosis in pancreatic cancer cells by downregulating PCAF proteins [347].

Furthermore, metformin exerts indirect effects on PDAC by reducing the levels of insulin and IGF-1 [348,349,350], as well as by modulating the gut microbiome [136,137]. It is important to note that the anti-PDAC effects of metformin are dose-dependent. Many studies investigating PDAC cells utilize higher concentrations of metformin (5–30 mM) compared to its physiological concentration (10–40 μM) [351,352,353,354]. Interestingly, low concentrations and high concentrations of metformin (>1 mM) suppress proliferation through AMPK-dependent and AMPK-independent pathways, respectively [355,356]. However, further research is necessary to fully comprehend the implications of metformin in PDAC progression and growth [340].

### 5.8. Prostate Cancer (PCa)

PCa is a significant cause of cancer-related mortality in the United States, accounting for one-third of all new cancer cases each year [357]. The primary treatment for PCa is androgen-deprivation therapy (ADT), but resistance to this therapy typically develops within 12–18 months, leading to the progression of the disease into castration-resistant prostate cancer (CRPC) and eventually the formation of metastases [358,359,360].

The relationship between diabetes, insulin levels, and the risk of prostate cancer is a topic of debate. Some studies have reported that patients with T2DM have a lower risk of developing PCa [361,362], while other researchers have presented conflicting findings [363,364,365]. Interestingly, elevated levels of insulin have been associated with the increased growth and mortality of PCa [366,367,368]. Additionally, high levels of IGF-IR have been linked to the invasion, aggressiveness, and poor prognosis of PCa [369,370,371,372,373].

Similarly, studies investigating the impact of metformin on PCa have yielded inconsistent results. Some studies have shown that diabetic patients who take metformin are less likely to develop PCa [217,374], but Zingales et al. [375] reported an increased risk of PCa development. However, Wu et al. and Chen et al. found no correlation between the use of metformin and the development of prostate cancer [376,377]. The combination of metformin with anti-PCa therapies has proven to be effective in reducing PCa cell proliferation and promoting cell death in laboratory settings both in vitro and in vivo [378]. This synergistic effect has been observed with the use of drugs such as bicalutamide [379], exenedin-4 [380], and 2-deoxyglucose (2DG) [381]. Interestingly, lower concentrations of metformin were found to be sufficient for it to exert its anti-PCa effects when combined with a Plk1 inhibitor [382], simvastatin [383], and solamargine [384]. Additionally, a recent study has revealed that vitamin D3 enhances the anticancer properties of metformin in PCa [385].

Several mechanisms have been described to explain the effects of metformin specifically in PCa. Metformin has been shown to induce cell cycle arrest at the G1/S phase in PCa cells by activating the AMPK/mTOR pathway [379]. In DU145 cells, LKB1 is required for metformin to activate AMPK, although it promotes cell death through a mechanism independent of LKB1-AMPK signaling [386]. Furthermore, the antiproliferative effects of metformin on PCa cells are mediated by REDD1 and cyclin D1 [97,378]. Metformin also leads to a significant reduction in the androgen receptor (AR) levels in LNCaP cells by decreasing c-MYC at both the protein and mRNA levels [387,388,389]. It is believed that metformin reverses the increase in AR expression by enhancing the activity of the MID1 translation regulator complex [375]. Moreover, metformin inhibits the proliferation of LNCaP cells by preventing the overexpression of androgen-dependent IGF-1R, primarily through the modulation of the mTORC1 complex rather than AMPK [390]. Similar to HCC, PCa is characterized by increased lipogenesis, which is associated with tumor growth and the development of aggressive forms of PCa [391]. Metformin has the ability to alter the expressions and activities of lipogenic enzymes and transcription factors, including FAS, ACC, and SREBP-1c, leading to energy depletion in cancer cells [392].

Multiple studies have demonstrated that androgen-deprivation therapy (ADT) can create microenvironments that are conducive to the development of hormone-independent cancer cells. This is achieved by increasing factors involved in the epithelial–mesenchymal transition (EMT) and exerting selective pressure towards the EMT [393,394]. Interestingly, metformin has been found to have various effects on the prostate cancer EMT. One mechanism is through the inhibition of the COX2/PGE2/STAT3 axis [395], which is known to be involved in the invasion and migration of prostate cancer cells [396,397,398,399]. In fact, metformin has been shown to sensitize castration-resistant prostate cancer (CRPC) patients to enzalutamide by blocking the EMT through this pathway [400]. Another mechanism is by reducing the expression of FoxM1 [401], a protein that plays a crucial role in cell proliferation, cell cycle regulation, angiogenesis, invasion, and metastasis [402,403]. Additionally, metformin inhibits the EMT by modulating microRNAs, such as miR30a and miR-708-5p, which have the ability to suppress tumor growth and metastasis [404,405,406]. Yang et al. demonstrated that metformin upregulates miR-708-5p in both LNCaP and PC3 cells, leading to the induction of endoplasmic reticulum stress and apoptosis [407].

## 6. Challenges of Metformin Repurposing as an Anticancer Drug

Numerous studies have examined the potential of metformin as an anticancer agent, revealing its ability to inhibit cancer proliferation, reduce the cancer risk, and enhance cancer prognoses. However, conflicting data and inconclusive results have also been reported, leading researchers to conduct further clinical trials to validate the beneficial effects of metformin in various types of cancer. A summary of ongoing trials can be found in Table 6. The diversity of findings can be attributed to several factors, including the study design [408,409]. Previous observational studies often suffered from residual confounding, selection bias, and immortal time bias, which cast doubts on their results [14,15,16,17,18]. Furthermore, variations in the analysis methodologies, the inclusion of different cancer types and subtypes, and differences in the patients’ demographics and diabetic conditions may have contributed to the inconsistent outcomes. Additionally, the diets followed by patients may also influence their responses to metformin. Elgendy et al. demonstrated that dietary restriction with intermittent fasting can improve the response to metformin [410].

The translation of in vitro studies into clinical settings is influenced by various factors, and one of them is the dosage of metformin used (Figure 3). In in vitro studies, it is common to use supraphysiological doses of metformin, typically ranging from 10 to 40 mM (330–6600 mg/L) and even up to 100 mM. These doses exceed the therapeutic plasma levels, which are typically between 0.465 and 2.5 mg/L [411,412]. The reason for using high doses in vitro is the non-physiologically high concentrations of the culture medium constituents, such as glucose, hormones, and growth factors [138]. However, such high doses are not feasible in clinical settings due to the potential for drug toxicity. It is important to note that the type of media used in cell cultures can also affect the cells’ sensitivity to metformin. For instance, cells cultured in DMEM require a higher dose of 10 mM of metformin to inhibit proliferation, whereas cells cultured in RPMI can be inhibited with a lower dose of metformin [413]. Even the varying concentrations of medium supplements, such as pyruvate and, to a less extent, glucose and aspartate, can render cells resistant to metformin [413].

Furthermore, the response to metformin may be influenced by the presence of metformin transporters. Metformin relies on transporters like OCT1, OCT3, and MATE1 to move across the cell’s plasma membrane [414,415]. The expressions of these transporters differ between normal cells and cancer cells, which raises uncertainty regarding the uptake of metformin by cancer cells. Additionally, the expressions of these transporters can be influenced by other medications. Shu et al. conducted a study that demonstrated how genetic variations in OCT1 can impact the effectiveness of metformin [416]. Consequently, the limited uptake of metformin by target cells may hinder its potential as a cancer treatment [417]. Moreover, several factors, including genetics, the microenvironment, the metabolic environment, biodistribution, and tissue specificity, can affect the sensitivity of cancer cells to metformin [418]. Notably, studies have highlighted the significance of p53, LKB1, and TSC2 in determining the responses of tumors to metformin [138].

Similarly, the prolonged use of metformin may diminish its effectiveness. Just like with chemotherapy, cancer cells have the ability to become resistant to metformin. It has been observed that MCF7 breast cancer cells, when subjected to long-term metformin therapy, developed a resistance not only to metformin but also to tamoxifen. This resistance was brought about by the activation of AKT-SNAIL1-E-Cadherin signaling [419]. Furthermore, Seo et al. have recently demonstrated that resistance to metformin could potentially contribute to the aggressive progression and spread of cancer [420]. These findings highlight the importance of considering these factors in future studies in order to determine the true anticancer effects of metformin.

Remarkably, the interest in repurposing metformin as an anticancer agent is gaining momentum for its low cost, as it has been off-patent since 2004 [421], its well-established safety profile, with side effects including mild–moderate GI discomfort and a metallic taste, which typically lessen over time, and the very rare incidence of lactic acidosis, compared to other biguanides [422]. Notably though, metformin use in males has been associated with an increased risk of genital malformations in male offspring [423]. This potential link is supported by in vitro studies demonstrating metformin’s effects on human and mouse testicular cells, potentially including testosterone suppression [424,425]. In addition, chronic use of metformin as well as high dosages (≥15 g/L) have been associated with vitamin B12 deficiency in 6–30% of patients due to malabsorption, as well as with alterations in the microbiota, motility, and calcium-dependent transport via the gastric intrinsic factor glycoprotein [426,427,428]. Yet, concurrent multivitamin use protects against vitamin B12 deficiency [429].

Despite its economic and clinical appeal for cancer therapy, repurposing metformin faces some regulatory hurdles. Conducting new clinical trials that adhere to rigorous protocols with well-defined patient populations, standardized regimens, and robust efficacy/safety data are needed to demonstrate the efficacy and safety against specific cancers. Re-evaluation of the safety profile is crucial, considering potential side effects in patients with co-morbidities and interactions with anticancer drugs. Additionally, the generic nature of metformin reduces financial incentives for large trials compared to patentable drugs, requiring justification based on existing treatment landscapes. Finally, intellectual property rights on metformin formulations specific to cancer treatment might require navigation. Thus, overcoming these challenges in clinical trial design, safety considerations, economic feasibility, and intellectual property management is key for metformin’s regulatory approval as an anticancer agent.

## 7. Conclusions

Metformin has emerged as a promising candidate for enhancing cancer treatment strategies due to its multifaceted approach in combating tumor formation and overcoming resistance to chemotherapy. The significant impact of cancer on global public health necessitates innovative therapeutic approaches, and the pleiotropic effects of metformin offer hope in addressing this urgent need. By targeting fundamental pathways involved in the development and progression of cancer, metformin provides a versatile tool for oncologists.

Observational studies and meta-analyses consistently demonstrate a decrease in the cancer occurrence and mortality among individuals treated with metformin, indicating its potential as a preventive agent. Furthermore, preclinical investigations have revealed its diverse mechanisms of action, ranging from metabolic modulation to direct interference with cancer cell growth and survival pathways. These findings highlight the wide range of anticancer effects exerted by metformin, which go beyond its primary use in managing T2DM.

However, the translation of promising preclinical results into clinical practice presents significant challenges. Conflicting outcomes from clinical trials emphasize the complexity of metformin’s interactions within the tumor microenvironment and the need for a nuanced understanding of its effects in different types of cancer and patient populations. Considerations regarding dosage, including the disparity between laboratory and in vivo concentrations, further complicate the therapeutic landscape. Additionally, the emergence of resistance to metformin underscores the importance of uncovering the underlying mechanisms and identifying strategies to mitigate or overcome this phenomenon.

Despite the obstacles faced, ongoing clinical trials provide optimism in terms of elucidating the role of metformin in cancer therapy and refining treatment strategies. The focus of biomarker discovery endeavors is to identify indicators that can predict the treatment response, thereby facilitating personalized approaches to the use of metformin in cancer patients. Additionally, efforts to optimize dosing regimens and explore combination therapies hold potential for enhancing the effectiveness of treatment and minimizing adverse effects.

To summarize, metformin presents a compelling avenue for advancing cancer treatment due to its multifaceted effects, which offer potential benefits across a wide range of malignancies. Although challenges persist, the continuation of research efforts holds promise in fully harnessing the therapeutic potential of metformin and improving outcomes for cancer patients worldwide. Through interdisciplinary collaboration and rigorous investigation, metformin may emerge as a fundamental component of modern cancer therapy, paving the way for more effective and personalized treatment approaches in the future.

## Figures and Tables

**Figure 1 ijms-25-04083-f001:**
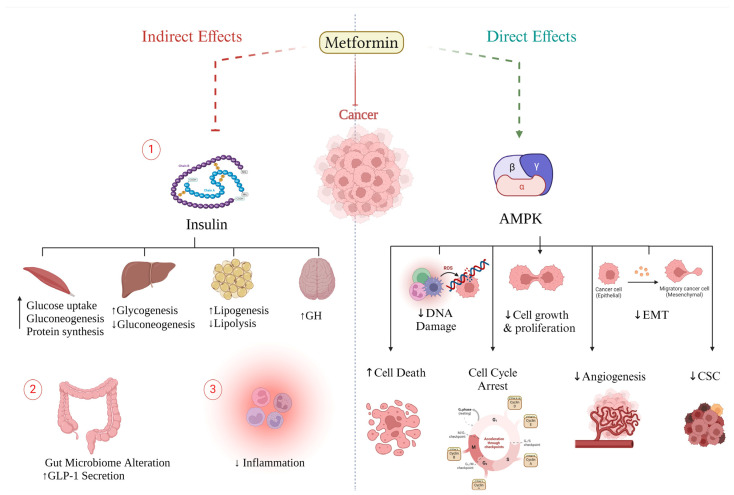
Summary illustration of the direct and indirect effects of metformin in cancer. GH: growth hormone; GLP-1: glucagon-like peptide 1; EMT: epithelial–mesenchymal transition; CSC: cancer stem cells. ↓: decrease; ↑: increase; ⊥: inhibits.

**Figure 2 ijms-25-04083-f002:**
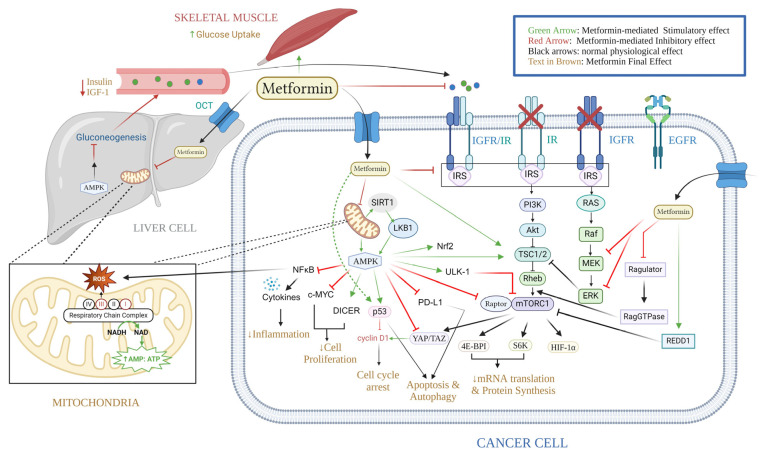
Detailed schematic illustration of anticancerous effect of metformin.

**Figure 3 ijms-25-04083-f003:**
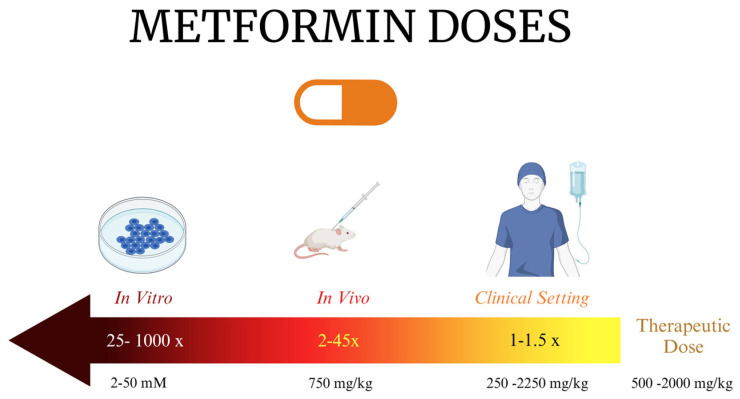
Differences in metformin dosing.

**Table 4 ijms-25-04083-t004:** Metformin impact on aging-related molecular and metabolic changes.

Aging Hallmark	Significance	Metformin Impact	
		At Molecular Level	Outcome
**Dysregulated nutrient signaling**	Distorted metabolic homeostasis	↓ IGF-1, ↓ mTOR ↑ AMPK, ↑ Sirt-1	Improved signaling
**Lost proteostasis**	Cell dysfunction and reduced viability	↑ UPR-related chaperone proteins: HSP60, HSP90, GRP78, and C/EBP ↑ KLF2	Suppressed protein misfolding and improved autophagy
**Mitochondrial dysfunction**	Energy and homeostasis are affected	↑ AMPK → H3K79m3 → ↑ SIRT3 → ↑ PGC-1α ↓ ROS production	Mitochondrial biogenesis upregulation and delayed aging
**Low-grade inflammation**	Epigenetic changes, no protein stability, and stem cell dysfunction	↑ Sirt-1 → ↓ NF-κB → ↓ cytokines (TNF-α & IL-6)	Inhibition of inflammation
**Telomere attrition**	Accelerated aging and diseases	AMPK regulates telomere transcription through telomere repeat RNA	Prevented telomere shortening
**DNA damage**	Genomic instability	↑ ATM, ↑ Checkpoint Kinases-2 ↓ AKT	Antioxidant effect and DNA damage prevention
**Stem cell exhaustion**	Impaired tissue regenerative capacity and dysfunction	↑ antioxidation, ↑ AMPK ↓ mTORC1 ↑ Nrf2 → ↑GPx7	Delayed premature cellular senescence and extended lifespan of stem cells
**Senescence and SASP**		↑ Sirt1 ↑ MBNL-1 → ↑miR-130a-3p → ↓ STAT3 ↓ NLRC4 ↓ NF-κB	Reduced premature senescence and SASP

MBNL-1: muscleblind-like 1. ↓: decreased; ↑: increased; →: causes

**Table 5 ijms-25-04083-t005:** Effect of metformin on miRNAs involved in BC proliferation.

MiRNA	Regulatory Effect	Effector Gene	Reference
miR-200c	↑	↓ AKT2	[229]
miR-21-5p	↓	↑ AMPK → ↓ mTOR	[230]
miR-27a	↓	↑ AMPKα2	[231]
miR-26a	↑	↓ PTEN, ↓ EZH2, ↓ BCL-2	[232]
miR-193b	↑	↓ FAS	[233]

↓: decrease; ↑: increase.

**Table 6 ijms-25-04083-t006:** Ongoing clinical trials assessing anticancer role of metformin in cancer.

Interventional Studies					
NCT Number	Conditions	Interventions	Outcome Measures	Phases	Location
NCT05759312	Ovarian clear-cell carcinoma	Zimberelimab	PFS, OS, DCR, duration of response, and recurrence pattern	Phase I and II	US
NCT04926155	Metastatic prostate cancer	ADT and abiraterone	PFS, OS, and radiographic PFS	Phase II	China
NCT04925063	Metastatic prostate cancer	ADT and abiraterone	Castration-resistant prostate cancer-free survival, OS, and radiographic PFS	Phase II	China
NCT05921942	Colorectal cancer	FOLFOX protocol	DCR, PFS, OS, and IL-6 levels	Phase III	Egypt
NCT03379909	Bladder cancer		ORR, time to recurrence, toxicity	Phase II	Netherlands
NCT06030622	Metastatic pancreatic cancer	Simvastatin and digoxin		Phase I	US
NCT01529593	Metastatic cancer refractory to standard therapy	Temsirolimus	Maximum Tolerated Dose (MTD) of temsirolimus and metformin and clinical tumor response	Phase I	US
NCT05929495	Glioblastoma	Temozolamide	PFS at 6 months post-treatment	Phase II	Italy
NCT02336087	Pancreatic adenocarcinoma	Gemcitabine hydrochloride/paclitaxel albumin-stabilized nanoparticles	Feasibility of, compliance with, and toxicity of combination, PFS, OS,	Phase I	US
NCT04758000	Osteosarcoma	Placebo	DFS and toxicity	Phase II	Italy
NCT04945148	Glioblastoma	Radiation IMRT, temozolomide	OS, ORR, PFS, safety, and tolerability	Phase II	Italy
NCT05445791	Non-small-cell lung cancer	Placebo	OS, ORR, PFS	RCT: Phase III	Mexico
NCT01042379	Breast cancer	Standard therapy	pCR, RFS, OS	RCT: Phase III	US
NCT05660083	HER2-negative breast cancer	L-NMMA	Define recommended dose, ORR, PFS	Phase II	US
NCT05023967	Breast cancer	Placebo	Frequency of DLT occurrence, ki67 changes, changes in the expressions of PP2A/GSK3β/MCL-1 axis and circulating biomarkers	RCT: Phase II	Italy
NCT05326984	Acute lymphoblastic leukemia	Prednisone; vincristine; doxorubicin; L-asparaginase; etoposide; cytarabine; methotrexate; and 6-mercaptopurine	Decrease in ABCB1 gene expression, increase in AMPK gene expression, OS	RCT	Mexico
NCT02186847	Lung cancer	Carboplatin and radiation therapy	PFS, OS, local–regional progression	Phase II	US
NCT01430351	Glioblastoma	Mefloquine/memantine hydrochloride/temozolomide	PFS and toxicity	Phase I	US
NCT05680662	Triple-negative breast cancer	Combination product: quercetin, EGCG, metformin, zinc	DFS at 3 and 10 yrs, toxicity	RCT: Early Phase I	
NCT05507398	Breast cancer	Placebo and atorvastatin		RCT: Phase IV	
NCT04248998	Triple-negative breast cancer	Preoperative chemotherapy	pCR, RFS, OS, AA and lipid profile modifications, concentration of insulin and IGF-I	RCT: Phase II	Italy
**Prospective Observational Studies**					
**NCT** **Number**	**Conditions**	**Interventions**	**Outcome Measures**		**Location**
NCT04947020	Rectal cancer		OS, DFS		Poland
NCT05192850	Recurrent endometrial carcinoma	Placebo	Endometrial cancer recurrence, PFS, OS		US
NCT04245644	Pancreatic cancer	Targeted drugs, such as aspirin, B-blockers, metformin, ACE inhibitors, statins	DFS, OS, pancreatic cancer progression		Italy

AA: amino acid; DCR: disease control rate; DFS: disease-free survival; ORR: overall response rate; OS: overall survival; pCR: pathologic complete response; PFS: progression-free survival; RFS: relapse-free survival.

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
