# Peer review of "Metformin: A Dual-Role Player in Cancer Treatment and Prevention"

_ijms, 2024, doi:10.3390/ijms25074083_

Round 1

Reviewer 1 Report

Comments and Suggestions for Authors

This review nicely summarizes the role of metformin in the treatment and prevention of cancer. However, the roles of metformin in increasing T cell lifespan and in addressing vitamin B12 deficiency could be valuable additions for readers.

Author Response

The impact of Metformin on T cells and immune system has been already discussed (line 192-198). The role of metformin use in vitamin B12 Deficiency has been added as possible metformin side effect (line 898-901).

Reviewer 2 Report

Comments and Suggestions for Authors

The review article prepared by Galal and colleagues is a comprehensive study that compiles consistent evidence of metformin application against cancers. The authors constructed a rational and interesting manuscript with scientific merit and worth publication in IJMS. Overall, no obvious flaws were detected; I have just a few suggestions to improve even more the quality of the document:

1) Include the description of the abbreviations in the fig 1; the current caption does not help understanding;

2) The authors could include the research strategy applied to select the studies; I believe that it is a scoping review, and a systematic methodology must be described for transparency (databases, keywords, inclusion, and exclusion criteria);

3) The potential limitations identified after reviewing all these studies should be included, as well as a brief paragraph of perspectives;

4) Table 3, line NCT01589367 [67] has a blank column. Please revise it.

5) Even considering the evidence supporting the use of metformin for preventing and treating cancer, regulatory aspects could be addressed to update the scientific community about the barriers to implementing its use. In the very humble opinion of this reviewer, it should be discussed in a brief paragraph.

Author Response

1) Include the description of the abbreviations in fig 1; the current caption does not help understanding

- Description of the abbreviations in fig 1 has been added.

2) The authors could include the research strategy applied to select the studies; I believe that it is a scoping review, and a systematic methodology must be described for transparency (databases, keywords, inclusion, and exclusion criteria)

- Research strategy applied to select the studies is now included in the manuscript (line 67-75)

3) The potential limitations identified after reviewing all these studies should be included, as well as a brief paragraph of perspectives

- Potential limitations identified have been added to the manuscript as suggested (Line 98-111).

4) Table 3, line NCT01589367 [67] has a blank column. Please revise it.

- Revised as requested.

5) Even considering the evidence supporting the use of metformin for preventing and treating cancer, regulatory aspects could be addressed to update the scientific community about the barriers to implementing its use. In the very humble opinion of this reviewer, it should be discussed in a brief paragraph.

- Regulatory aspects are now addressed briefly as suggested (line 902-913).

Reviewer 3 Report

Comments and Suggestions for Authors

This manuscript sheds light on the potential use of metformin as an anti-cancer agent. It highlights the various mechanisms of action of metformin, particularly its ability to activate AMPK and inhibit mTOR. The authors also takes a closer look at the results of several studies that have explored the effects of metformin on various cancer types. After analyzing these studies, authors conclude that metformin could be a promising option for cancer prevention and treatment. Authors also acknowledge the presence of inconsistencies between these studies, emphasizing the need for further research. The manuscript proposes a possible explanation for how metformin could impact cancer development by altering factors such as diabetes, aging, hyperlipidemia, obesity, and inflammation. The manuscript is clear and well written - I have only some minor question:
Authors report some inconsistent findings regarding metformin's effects on specific cancer types - what factors could contribute to these inconsistencies, and what are the reasons for discrepancies in some clinical trials that failed to confirm the anti-cancer role of Metformin?
What are the potential side effects of Metformin when used as an anti-cancer agent?
What are the potential new targets for Metformin in the discovery of anticancer drugs?

Author Response

1) Authors report some inconsistent findings regarding metformin's effects on specific cancer types – what factors could contribute to these inconsistencies, and what are the reasons for discrepancies in some clinical trials that failed to confirm the anti-cancer role of Metformin?

- All factors contributing to the discrepancies between the studies have already been discussed thoroughly in section 6 (lines 837-887) and the limitations of the studies are now added additionally (lines 98-111).

2) What are the potential side effects of Metformin when used as an anti-cancer agent?

- Potential side effects of Metformin have been added as suggested (lines 889-901).

3) What are the potential new targets for Metformin in the discovery of anticancer drugs?

- The following are the potential target of metformin mentioned in the article

  • Cancer stem cells (line 255-257, 333-334, 603-604)
  • Angiogenesis (line 447-454)
  • Immunotherapy (line 192-198)

However, to avoid any confusion to the readers, the sentence has been omitted from the introduction.